# RttA, a Zn$_2$-Cys$_6$ transcription factor in *Aspergillus fumigatus*, contributes to azole resistance

Takahito Toyotome,[1,2,3,4] Hiroki Takahashi,[4] Akira Watanabe,[4] Daisuke Hagiwara[4,5,6]

**ABSTRACT**  *Aspergillus fumigatus* is a common environmental fungus and the leading cause of aspergillosis, an opportunistic infection in humans and animals. The mortality rate of aspergillosis remains high despite antifungal treatments, with azole antifungals, such as voriconazole, being the primary treatment. However, resistance to azoles is increasing, partly because of environmental exposure to agricultural fungicides. In a previous study, we identified the protein RttA to be involved in azole susceptibility. To further understand the role of RttA in azole resistance, we conducted RNA-Seq analysis and functional analyses by constructing RttA deletion and overexpression strains. The transcriptome data revealed that RttA contains a Zn$_2$-Cys$_6$ domain and may function as a transcription factor. Furthermore, six genes, including a γ-glutamyl transpeptidase gene (*ggtA*) and a homolog of factor C (*facC*), were shown to be putatively regulated by RttA. These findings suggest that RttA is involved in azole resistance by regulating several genes, highlighting its potential as a target for developing antifungal strategies against *A. fumigatus*.

**IMPORTANCE**  This study identified RttA as a Zn$_2$-Cys$_6$ transcription factor in *Aspergillus fumigatus*, a critical pathogen that causes aspergillosis. RttA influences azole susceptibility. Our results demonstrated that the deletion of RttA increases susceptibility to azoles, whereas its overexpression enhances resistance. This discovery is crucial because the mechanism underlying azole resistance has not yet been fully elucidated. Hence, understanding the role of RttA provides insights into resistance mechanisms and potential targets for new therapeutic strategies, thereby addressing the urgent need for effective treatments against azole-resistant *A. fumigatus* strains.

**KEYWORDS**  azole, tolerance, *Aspergillus fumigatus*, rttA gene, zn2-Cys6 transcription factor

*A*spergillus fumigatus is a ubiquitous saprophytic fungal species in the environment, including the soil and air. It is recognized as the most common causative agent of aspergillosis, an opportunistic infectious disease in humans and animals. The clinical presentation of aspergillosis varies and includes acute, chronic, and allergic bronchopulmonary aspergillosis. The primary target organ of the pathogen is the lungs; however, depending on the host condition, *A. fumigatus* can be disseminated to other organs, such as the brain, heart, and skin. An estimate showed that the number of patients affected by invasive aspergillosis and chronic pulmonary aspergillosis was more than one million and three million, respectively (https://gaffi.org/why/fungal-disease-frequency/, "Fungal Disease Frequency"; accessed on December 6, 2024). The mortality rate of aspergillosis remains high even with antifungal treatments (1). Moreover, the World Health Organization (WHO) has classified *A. fumigatus* as a pathogen in the Critical Priority Group (2).

Address correspondence to Takahito Toyotome, toyotome-takahito-yu@iuhw.ac.jp.

The authors declare no conflict of interest.

See the funding table on p. 11.

Azole antifungal agents such as voriconazole are the first-choice drugs for aspergillosis. Azole antifungal agents inhibit lanosterol 14-α-demethylase Cyp51A (or Erg11), which catalyzes a step in ergosterol biosynthesis. Azole resistance in *A. fumigatus* (AR*Af*) occurs in environments where fungicides are used in agricultural fields and in clinical settings where medical antifungals are used in patients during long-term azole therapy (3). In particular, environmentally acquired AR*Af* is becoming a global concern because there is evidence that AR*Af* spreads across borders via agricultural products, such as floral plant bulbs (4–6). AR*Af* is recognized as a drug resistance threat with the potential to spread or become a challenge in the United States and is listed on the Watch List by the US Centers for Disease Control (7). The major mechanism for acquiring azole resistance involves qualitative and/or quantitative changes in Cyp51A. Most environmentally acquired AR*Af* contain 34- or 46-base pair tandem repeats in the promoter region, leading to increased *cyp51A* expression and nonsynonymous mutations, leading to a qualitative change in the coding enzyme Cyp51A. Conversely, the AR*Af* strains that emerge during prolonged azole treatment occasionally develop mutations in several genes associated with azole susceptibility. Although such mutations have been reported in the azole-targeting enzyme Cyp51A, mutations affecting azole susceptibility have also been identified in other proteins, including Hmg1 (8, 9) and HapE (10). Our previous study explored alternative genes responsible for azole susceptibility *in vitro* using the commonly used agricultural azole fungicide tebuconazole. RttA (gene ID: AFUA_7G04740 and AFUB_090280) was later identified as a protein responsible for azole susceptibility (11). Recently, the expression of *rttA* was found to be higher in voriconazole persisters, suggesting that this gene is important in the presence of voriconazole (12). An ortholog of RttA is present in other *Aspergillus* species; however, its mechanisms associated with azole susceptibility and its functions remain unknown.

To gain more insights into the role of RttA in azole resistance, we conducted functional analysis in this study by constructing RttA deletion mutants and overexpression strains. RNA sequencing (RNA-Seq) data revealed that the previous prediction of *rttA* gene structure was incorrect and that RttA contains a $Zn_2$-$Cys_6$ domain and may function as a transcription factor. Furthermore, azole susceptibility assays using the RttA deletion and overexpression strains showed that RttA contributed to azole resistance. Furthermore, six putative RttA-regulated genes were identified via transcriptomic analysis. In the upstream regions of these genes, putative binding motifs of RttA were predicted. These findings suggest that RttA is involved in azole resistance by regulating several genes. In addition, we have elucidated in more detail the function of RttA, which has remained unknown until now, and highlighted its potential as a target for developing antifungal strategies against *A. fumigatus*.

## MATERIALS AND METHODS

### Strains and plasmids

The *Aspergillus fumigatus* A1159 (AfS35) and pSK485 plasmids used in this study were obtained from the Fungal Genetics Stock Center (FGSC), Kansas State University (Manhattan, KS, USA). *akuA* was deleted in the A1159 strain. *Escherichia coli* DH5α was used as the plasmid host. The plasmids and primers used in this study are listed in Table S1 and S2, respectively.

### Total RNA preparation from *Aspergillus fumigatus*

Total RNA for RNA-Seq from *A. fumigatus* was prepared using a Direct-zol RNA Mini-Prep Kit and TRI-Reagent (Zymo Research Corporation, Irvine, CA, USA), with slight modifications. Briefly, $2.5 \times 10^4$ spores of *A. fumigatus* strains were inoculated to 2.5 mL of RPMI-1640 medium (R8755, Merck KGaA, Darmstadt, Germany) buffered with 0.165 mol/L MOPS (pH 7.0) in a six-well cell culture plate (Nippon Genetics Co., Ltd., Tokyo, Japan) and cultivated for 24 h at 35°C or for 48 h at 37°C under 5% $CO_2$. Hyphae

in each well were scraped and placed in a 2 mL tube with 1.0 mm zirconia beads (ZB-10, TOMY Seiko Co., Ltd., Tokyo, Japan). Fungal cells were freeze-dried using a freeze-dryer (FDU-2200; Tokyo Rikakikai, Co., Ltd., Tokyo, Japan) and then crushed for 5 min using a Vortex-Genie 2 (Scientific Industries, Inc., Bohemia, NY, USA). TRI-reagent (1 mL) was then added to the crushed fungal cells, the liquids were mixed via inversion, and 0.2 mL chloroform was added to the lysate. After centrifugation, the upper phase was transferred to a new 1.5 mL tube, and an equal volume of ethanol was added to the solution. The procedure was performed according to the manufacturer's instructions (Zymo Research Corporation). The RNA quantity and quality were determined using a NanoDrop 1000 (Thermo Fisher Scientific, Waltham, MA, USA) and an Agilent RNA 600 Pico Kit with an Agilent 2100 Bioanalyzer (Agilent Technologies, Inc., Santa Clara, CA, USA), respectively.

## RNA sequencing

The total RNA sample of *A. fumigatus* A1159 used to confirm the coding region of *rttA* was prepared from a culture at 35°C for 24 h and sent to the Bioengineering Lab Co., Ltd. (Kanagawa, Japan), which prepared the library and performed sequencing. An MGIEasy RNA Directional Library Prep Set (MGI Tech Co., Ltd., Shenzhen, China) was used for library preparation. The quantity and quality of the prepared library were assessed using a QuantiFluor dsDNA System (Promega Corporation, Madison, WI, USA), BioTek Synergy H1 (Agilent Technologies, Inc.), dsDNA 915 Reagent Kit (Agilent Technologies, Inc.), and Fragment Analyzer (Agilent Technologies, Inc.). For sequencing using the DNBSEQ-G400RS High-throughput Sequencing Kit (MGI Tech Co., Ltd.), the DNA library was circularized using an MGIEasy Circularization Kit (MGI Tech Co., Ltd.). The circularized library was then sequenced under $2 \times 100$ bp conditions using a DNBSEQ-G400 system.

The total RNA samples of A1159 (parental strain), $\Delta rttA$, and *rttA*-overexpressing (*rttA*OE) strains were prepared from 48 h cultures at 37°C under 5% $CO_2$ and analyzed using a Novaseq X Plus platform (Illumina, Inc., San Diego, CA, USA) by Novogene Co., Ltd. (Beijing, China) for RNA-Seq analysis. Three total RNA samples from each strain were prepared and analyzed. The number of raw reads and the quality of the obtained data are summarized in Table S3.

## Mapping RNA-Seq data on *A. fumigatus* genome

To confirm the coding region of *rttA*, the obtained sequences from *A. fumigatus* A1159 strain were processed using Cutadapt software (ver. 4.0) (13) to remove adaptor sequences and sickles (ver. 1.33) (14) to remove low-quality reads (the parameters in sickle were "-q 20 -L 40"). The filtered reads were then mapped to an *A. fumigatus* A1163 genome assembly (GenBank accession number GCA_000150145.1) using HISAT2 (ver. 2.2.1) (15). The produced sam-formatted data were transformed into sorted indexed BAM-format data using samtools (ver. 1.16.1) (16). Finally, the data were visualized using Integrative Genomics Viewer (IGV, ver. 2.16.1 and 2.17.4) (17).

The mapping of data obtained from the A1159, $\Delta rttA$, and *rttA*OE strains was performed by Novogene Co. Ltd. using HISAT2 (18). The mapping statistics are presented in Table S4.

## RT-PCR and sequencing

A PrimeScript One Step RT-PCR Kit Ver.2 (Takara Bio Inc., Shiga, Japan) was used for reverse transcription and subsequent PCR. The primers used to confirm the cDNA sequence of *rttA* are listed in Table S2. Sanger sequencing was performed by Eurofins Genomics (Tokyo, Japan).

## Antifungal susceptibility test

A broth dilution method based on CLSI M38-3Ed (19) was performed to determine the minimum effective concentrations (MECs) of micafungin (MCFG) and caspofungin (CPFG), and the minimum inhibitory concentrations (MICs) of amphotericin B (AMPH-B),

voriconazole (VRCZ), and itraconazole (ITCZ) using an Eiken Dry Plate (Eiken Chemical Co., Ltd., Tokyo, Japan). The MECs and MICs were determined 48 h after inoculation. The MIC distributions of the ΔrttA mutant and the rttAOE overexpression strain were compared using the Mann–Whitney U test. A P-value of less than 0.05 was considered statistically significant.

## Plasmid constructions

The plasmids pSK485-hygro and pSK485-hygro-PgpdA were constructed via the Gibson assembly method using a Gibson Assembly Master Mix (New England Biolabs, Ipswich, MA) according to the manufacturer's instructions. A KOD One PCR Master Mix (Toyobo Co., Ltd., Osaka, Japan) was used to obtain fragments for assembly.

## β-rec/six-blaster cassette and CRISPR RNA preparation for genome editing in A. fumigatus

A β-rec/six-blaster cassette with 35 bp overhangs in both ends for genome editing in A. fumigatus was prepared via PCR using a KOD One PCR Master Mix (Toyobo Co., Ltd.). pSK485-hygro and pSK485-hygro-PgpdA were constructed as the cassette templates in this study. The resulting amplicons were then purified using a QIAquick PCR Purification Kit (QIAGEN N. V., Venlo, Netherlands). The sequences of the CRISPR RNAs (crRNAs) used in this study are listed in Table S5.

## Protoplasting and transformation using a CRISPR/Cas9 system

Protoplasts of A. fumigatus were prepared as described previously (20, 21). Briefly, A. fumigatus A1159 conidia were inoculated in yeast extract-glucose medium (20) and cultured for 6 h at 37°C. After cultivation, VinoTaste Pro (Novozymes A/S, Bagsværd, Denmark) was added to the germling suspension at a final concentration of 0.05 g/ml. After 1 h of cultivation at 30 or 37°C, the protoplasts were recovered via centrifugation. After rinsing with a solution containing 0.6 M KCl and 50 mM $CaCl_2$, the protoplasts were suspended in 900 µL of a solution containing 0.6 M KCl and 50 mM $CaCl_2$. Next, 100 µL of the resultant suspension was used for transformation, which was performed as previously described (22). Trans-activating crRNA (tracrRNA) and Cas9 were purchased from IDT (Newark, NJ, USA). The ribonucleoprotein complex of crRNA, tracrRNA, and Cas9 protein (RNP) and a PCR fragment of the β-rec/six-blaster cassette with 35 bp overhangs in both ends were mixed and treated with polyethylene glycol. The transformants were then plated on Aspergillus minimal medium (AMM) agar supplemented with 1.2 M sucrose. After overnight cultivation, AMM supplemented with approximately 0.5% agar and 400 µg/mL hygromycin B (FUJIFILM Wako Pure Chemical Corporation, Osaka, Japan) was overlaid, and the plates were incubated until colonies appeared. The colonies were plated twice on AMM supplemented with xylose (0.5%) to remove the β-rec/six-blaster cassette. The elimination of the cassette and the desired construction were confirmed via PCR. The resulting strains were used for further experiments.

## RttA 3D structure prediction and comparison against data deposited in the Protein Data Bank

The 3D structure of RttA was predicted using AlphaFold Monomer ver. 2.0 (23) for Benchling (24). The predicted data were queried against the data deposited in the Protein Data Bank of DALI (25).

## Differential expression and pathway analysis

iDEP 0.96 was used for principal component analysis (PCA), differential gene expression, and pathway analyses (26). DESeq2 was used with the default settings to identify differentially expressed genes (DEGs).

## UPC2 motif search

JASPAR 2024 was used to search for the UPC2-binding motif (matrix profiles MA.0411.1 UPC2 and MA.0411.2 UPC2 in JASPAR 2024). Target genes were extracted using DESeq2 analysis as described above. The putative transcription start sites (TSS) were determined based on the mapping data of RNA reads, and 1 kb upstream regions from the TSS were used in the analysis. Predicted motifs with relative scores ≥ 0.95 are listed in Table S6-S11.

## RESULTS

### Correction of a genetic prediction in the *rttA* gene

No functional domains in the Pfam database were detected in the *rttA* gene (AFUA_7G04740 and AFUB_090280) of *A. fumigatus*. However, some syntenic homologs of other *Aspergillus* spp. listed in FungiDB (27) possess a $Zn_2$-$Cys_6$ fungal-type DNA-binding domain (11). An RNA-Seq experiment was then performed to confirm the construction of the rttA transcript of *A. fumigatus*. The mapped reads in the *rttA* gene and its upstream region were analyzed (Fig. 1A). The results showed that an intron was overlooked in the upstream region of the *rttA* gene annotated in the genome data (Fig. 1A and B). The re-predicted gene was found to contain a GAL4-like $Zn_2$-$Cys_6$ fungal-type DNA-binding domain (domain architecture ID 10637797) in the N-terminal region following an NCBI Conserved Domain Search (28). Six conserved cysteines in this domain were found in the RttA of *A. fumigatus* (Fig. 1B). Moreover, a small exon was detected via RNA-Seq analysis between the previously annotated exons 1 and 2 of *rttA* (Fig. 1A and Fig. S1). The upstream and extra-small introns were confirmed via RT-PCR and Sanger sequencing. These results indicated that RttA has an overlooked $Zn_2$-$Cys_6$ fungal-type DNA-binding domain and a small overlooked exon. Hereafter, genes and proteins containing the $Zn_2$-$Cys_6$ fungal-type DNA-binding domain are referred to as *rttA* and RttA, respectively.

The structure of RttA was predicted using the AlphaFold Monomer ver. 2.0 on Benchling, and the data were analyzed using DALI (29). The output results (Supplementary Data 1) showed that the UPC2 of *Saccharomyces cerevisiae* had a structure most similar to that of RttA. Moreover, 20 of the 42 amino acids in the sequences of their DNA-binding domains (positions 9 to 50 in RttA and 46 to 87 in UPC2) are identical.

### RttA plays a role in azole resistance

To gain more insights into the function of RttA in azole resistance, we constructed *rttA* deletion mutant and overexpression strains (Fig. S2). The antifungal susceptibilities of the impaired strain (Δ*rttA*) and the rttA-overexpressing strain (*rttA*OE) were determined using the broth microdilution method (Table 1). The Δ*rttA* strain showed increased susceptibility to ITCZ and VRCZ compared to the parental strain A1159. By contrast, the *rttA*OE strain showed decreased susceptibility to VRCZ compared to the parental strain A1159. These data indicate that RttA contributes to azole resistance.

### RttA upregulates the expression of six genes

To identify which genes were associated with RttA expression, affecting susceptibility to azole, we conducted RNA-Seq using the Δ*rttA* and *rttA*OE strains. Among three strains, including the parental strain, DESeq2 of iDEP 0.96 showed DEGs among all comparisons (Fig. 2). PCA was also performed using iDEP 0.96 on transcriptomic data from the three strains. The resulting PCA plot (Fig. S3) showed a clear separation among the three sample groups, indicating distinct transcriptomic profiles and suggesting that the biological differences are robust and reproducible across biological replicates. Δ*rttA* strain exhibited 53 upregulated genes compared to the *rttA*OE strain. Conversely, the *rttA*OE strain showed 69 upregulated genes compared to the Δ*rttA* strain. The *rttA*OE strain showed 55 downregulated genes compared to the A1159 parental strain. The expression of 155 genes was increased in *rttA*OE compared to the parental strain. The

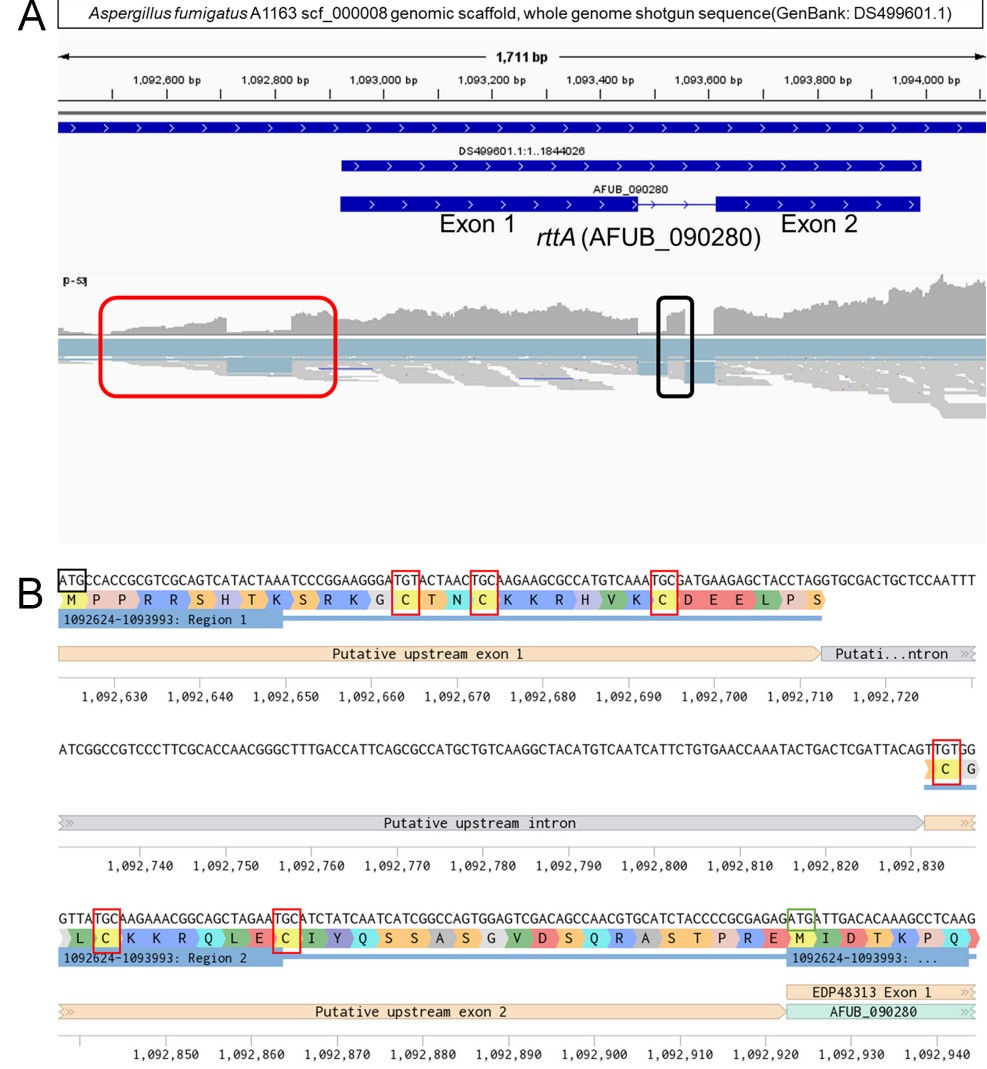

**FIG 1** The overlooked exons are located in the upstream region, and an intron was found between exons 1 and 2 of *rttA*. (A) Diagram of the RttA-coding region and the flanking regions of the scf_000008 genomic scaffold of *A. fumigatus A1163* (GenBank: DS499601.1). The image is a screenshot captured from IGV software. The upper part indicates the location of *rttA* (AFUB_090280) in the scf_000008 genomic scaffold and the region including *rttA*. The lower part shows the mapped reads obtained via RNA-Seq analysis. The red box indicates a large overlooked intron and flanking exons in the upstream region of *rttA*. The black box indicates a small overlooked exon between exon 1 and exon 2 of *rttA*. (B) The overlooked N-terminal region of the RttA protein contains a $Zn_2$-$Cys_6$ zinc cluster. The image is a screenshot created using Benchling (24). The ATG in a black box indicates the putative start codon of the RttA protein. The ATG in a green box indicates the start codon of the RttA previously annotated. The six red boxes indicate the cysteines comprising the $Zn_2$-$Cys_6$ zinc cluster. The axis indicates the nucleotide position in the scf_000008 scaffold.

*ΔrttA* strain exhibited fewer DEGs than the A1159 strain, with five genes downregulated. The *ΔrttA* strain showed an increased expression of 80 genes compared to the A1159 parental strain.

Next, we focused on genes whose expression increased at least two-fold in the *rttA*OE strain compared to both the A1159 parental and *ΔrttA* strains and increased in the wild-type strain compared to *ΔrttA*. Under these conditions, seven genes, including *rttA*, were identified (Tables 2 and 3). Three genes, whose expression increased at least twofold in the *rttA*OE strain compared to both the A1159 parental strain and the *ΔrttA* strain, and increased at least twofold in the wild-type strain compared to the *ΔrttA*, were extracted (Fig. 2B).

**TABLE 1** Susceptibility of A1159, *ΔrttA*, and *rttA*OE strains to antifungal drugs[b]

| | MEC (µg/mL) | | MIC (µg/mL) | | |
|---|---|---|---|---|---|
| | **MCFG** | **CPFG** | **AMPH-B** | **ITCZ** | **VRCZ** |
| A1159 | ≤0.015 | 0.125 | 1 | 0.25 | 0.5 |
| *ΔrttA*[c] | ≤0.015 (≤0.015) | 0.125 (0.125) | 0.758 (0.5–1) | 0.165 (0.125–0.25) | 0.287[a] (0.25–0.5) |
| *rttA*OE[c] | ≤0.015 (≤0.015) | 0.144 (0.125–0.25) | 1.32 (1-2) | 0.287 (0.25–0.5) | 1.149[a] (0.5–2) |

[a]Values marked with the symbol "a" are significantly different from each other ($P < 0.05$, Mann–Whitney U test).
[b]MEC, minimum effective concentration; MIC, minimum inhibitory concentration; MCFG, micafungin; CPFG, caspofungin; AMPH-B, amphotericin B; ITCZ, itraconazole; VRCZ, voriconazole.
[c]The data from this strain are the geometric means of MIC values obtained from five independent colonies. The figures in parentheses indicate the range of MIC values.

AFUB_090290 and AFUB_090300 are located downstream of *rttA* (AFUB_090280). Their expressions in *rttA*OE increased 20.6-fold and 18.3-fold, respectively, compared to the *ΔrttA* strain. Among the 54 upregulated genes in the *rttA*OE strain compared to the A1159 parental strain and *ΔrttA* strain (Fig. 2B), the expression of four genes was increased in the A1159 strain compared to the *ΔrttA* strain (Table 2). The expression of the AFUB_049010 (*facC*) gene, which encodes a FacC-like secreted protein, increased 31.1-fold and 48.8-fold in the *rttA*OE strain compared to the A1159 and *ΔrttA* strains, respectively. The expressions of AFUB_046570, AFUB_099610, and AFUB_094790 in the *rttA*OE strain were at least two times higher than those in the *ΔrttA* and A1159 strains (Table 2). When comparing the expression levels of these four genes, no more than a twofold change was observed between the A1159 and *ΔrttA* strains (Table 2).

## An extended N-terminal region of γ-glutamyl transpeptidase was found in the AFUB_090290-encoded region

The RNA-Seq data showed that AFUB_090290 and AFUB_090300 were upregulated in the RttA-overexpressing strain. However, the mapped reads were found in the intergenic region between them (Fig. S4). Furthermore, based on the RNA-mapping data, two introns and two exons were deduced to be present in the upstream region of AFUB_090300 (Fig. S4). Via manual reannotation, this region was presumed to encode a putative γ-glutamyl transpeptidase (*ggtA*) gene. A signal peptide cleaved between Thr27 and Ser28 was predicted in the N-terminal region of this putative γ-glutamyl transpeptidase gene using SignalP 6.0 (30).

## UPC2-binding motifs were found in the upstream regions of the genes upregulated by RttA

As shown above, RttA is homologous to UPC2; therefore, the binding motif of UPC2 was searched in the upstream regions of each target gene. Initially, RttA was analyzed to determine the possibility of autoregulation. As shown in Table S6, two regions whose sequences were similar to the UPC2-binding motif were found in the upstream region of *rttA*. The re-predicted putative γ-glutamyl transpeptidase gene (*ggtA*) described above possesses a promoter region in the *rttA* gene. This region contained two sequences similar to UPC2-binding sites, with relative scores > 0.95 (Table S7). Five putative UPC2-binding motifs in the 1 kb upstream region of *facC* were then identified (Table S8), while one to three motifs in each 1 kb upstream region of AFUB_046570, AFUB_099610, and AFUB_094790 were predicted to be putative UPC2-binding sites (Table S9-11). These data suggest that these genes are regulated through the binding motifs of the UPC2 homolog RttA.

## DISCUSSION

Our previous study demonstrated that mutations that accumulate in RttA resulted in azole resistance (11). However, the function of RttA remained unknown because no characteristic domains have been identified in the *rttA* gene from annotated data on *A. fumigatus* Af293 (GCA_000002655.1) and A1163 genomes (GCA_000150145.1). In this

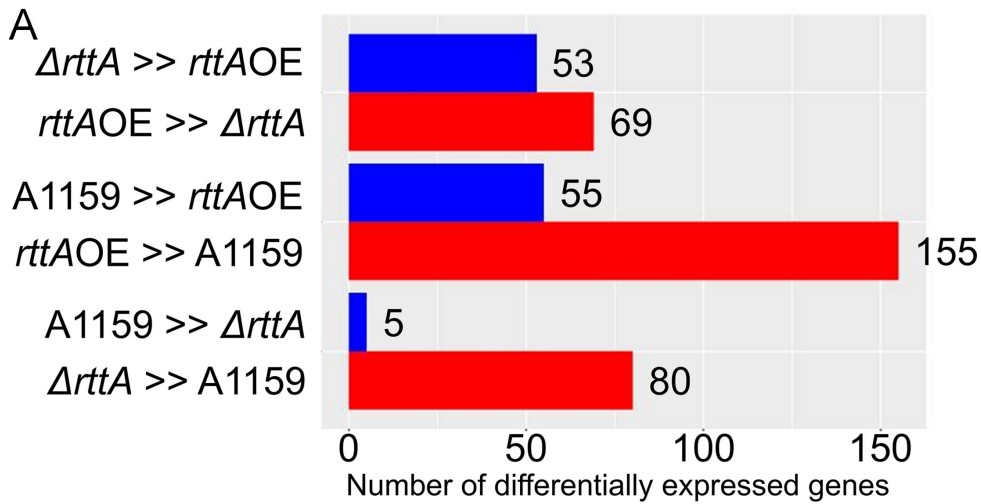

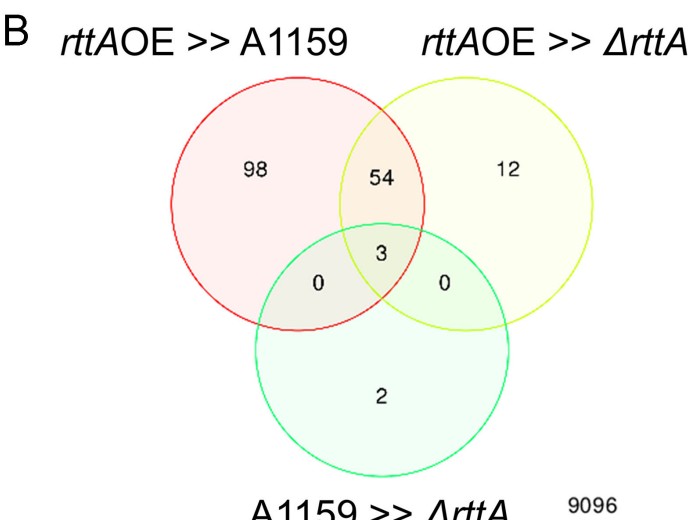

**FIG 2** Summary of the number of genes with significantly altered expression levels (minimum fold change cutoff and FDR cutoff are 2 and 0.1, respectively) among the A1159, ΔrttA, and rttAOE strains. (A) The number of genes with significantly altered expression levels between paired comparisons is shown on the left side of the graph. (B) Comparisons of the gene expression levels among three strains.

study, we reannotated RttA as a putative transcriptional factor that contains a $Zn_2$-$Cys_6$ fungal-type DNA-binding domain in its N-terminal region. Indeed, a six-Cys motif is present in the $Zn_2$-$Cys_6$ zinc cluster, which is involved in zinc-dependent DNA binding. The most prominent protein containing the $Zn_2$-$Cys_6$ domain in *Aspergillus* species is AtrR, which plays a key role in the azole resistance of *Aspergillus* species, including *A. fumigatus*, by binding to DNA in the upstream region of *cyp51A* and *cdr1B* (31, 32). In addition, a small exon was found in the *rttA* gene between exons 1 and 2, which corresponded to ortholog sequences in *A. nidulans* FGSC A4 and *A. parasiticus* CBS 117618. This shows that the structure of genes may be mispredicted in public databases and that manual curation for transcripts may be necessary before an accurate functional analysis of genes of interest can be performed.

    Next, we demonstrated that ΔrttA was more susceptible to VRCZ than the parental strain. By contrast, the rttAOE strain was more resistant to VRCZ. These data indicate that putative RttA-regulated genes contribute to VRCZ resistance. We then compared the transcriptomes of the parental strain (A1159), ΔrttA, and rttAOE, revealing that

**TABLE 2** Fold changes of seven gene expressions compared among A1159, ΔrttA, and rttAOE

| Gene ID | rttAOE vs ΔrttA[a] | | rttAOE vs A1159[a] | | A1159 vs ΔrttA[a] | |
|---|---|---|---|---|---|---|
| | Log₂-fold change | Adjusted p-value | Log₂-fold change | Adjusted p-value | Log₂-fold change | Adjusted p-value |
| AFUB_090280 (rttA) | 10.61473181 | 1.66E-244 | 2.67014081 | 1.88E-161 | 7.944591 | 2.06E-135 |
| AFUB_049010 (facC) | 5.608812747 | 0 | 4.95840451 | 0 | 0.650408237 | 0.001991099 |
| AFUB_090290 | 4.363443907 | 0 | 2.685315826 | 5.69E-150 | 1.678128081 | 7.96E-45 |
| AFUB_090300 | 4.194663674 | 0 | 2.616333934 | 1.60E-199 | 1.57832974 | 6.70E-67 |
| AFUB_046570 (aihA) | 2.892523604 | 1.36E-100 | 2.297776945 | 8.24E-66 | 0.594746659 | 0.000397569 |
| AFUB_099610 | 2.820353973 | 2.15E-205 | 2.47769507 | 5.19E-169 | 0.342658903 | 0.006725243 |
| AFUB_094790 | 1.306320809 | 6.90E-42 | 1.009789034 | 1.50E-25 | 0.296531775 | 0.020520126 |

[a]The values of "A vs B" are expressed as the ratio of expression levels in strain A to those in strain B.

the expression of five genes (*ggtA*, *facC*, *aihA* [AFUB_046570], AFUB_099610, and AFUB_094790) was regulated by RttA.

Although AFUB_090300 partially covered the GgtA-homologous region, an extended N-terminal region containing a putative signal peptide was found in this study, indicating that it is a secreted protein. Although limited information on fungal γ-glutamyl transpeptidases is available, bacterial γ-glutamyl transpeptidases have been characterized extensively (33). A phylogenetic analysis by Verma (34) revealed that eukaryotic γ-glutamyl transpeptidases, including fungal γ-glutamyl transpeptidases, were closely related to γ-glutamyl transpeptidases of bacteria known to be human pathogens. Other fungal γ-glutamyl transpeptidases, such as those from *A. sydowii* and *A. oryzae*, were characterized through their enzymatic properties, including their substrates (35, 36). Following an alignment of GgtA sequences from *A. fumigatus*, *A. sydowii*, and *A. oryzae*, the pairwise identity rates of GgtA of *A. fumigatus* compared to those of *A. sydowii* and *A. oryzae* are 71.8% (mismatched residues: 167/593 amino acids) and 75.7% (mismatch: 144/584 amino acids), respectively; however, the substrate-binding residue and cysteines forming a disulfide bond (36) are conserved among the three GgtAs. Several imidazole compounds are γ-glutamyl acceptors of γ-glutamyl transpeptidases (35, 36). Azoles, including VRCZ, contain a triazole, a related heterocyclic structure. Further analyses are needed to reveal whether azole compounds can be modified by γ-glutamyl transpeptidases.

FacC (AFUB_049010) in *A. fumigatus* is a homolog of factor C in *Streptomyces griseus*, which is currently identified as *S. albidoflavus* (37). Factor C was previously detected and isolated from the culture fluid of *S. griseus* 45H (38, 39) and is known to induce sporulation (40, 41). Factor C from *S. griseus* 45H also affects transcription (42) and K⁺ release (43). FacC is also known as a homolog of TagC in *Bacillus subtilis*. Its putative function is predicted to be polyglycerol phosphate assembly and the export of proteins involved

**TABLE 3** A summary of six genes, including *rttA*

| Gene symbol | Gene ID | |
|---|---|---|
| *rttA* | AFUB_090280 | Zn₂-Cys₆-type transcription factor. It has an extended N-terminal region and contains a Zn₂-Cys₆ domain in this study. |
| *facC* | AFUB_049010 | It is a homolog of teichoic acid biosynthesis protein C (FacC). A signal peptide was predicted in the protein. |
| *ggtA* | AFUB_090290 AFUB_090300 | A putative γ-glutamyltranspeptidase. Their regions encode a single polypeptide containing a signal peptide. |
| *aihA* | AFUB_046570 | It has a domain with predicted protein-arginine deiminase activity and role in the putrescine biosynthetic process.[a] |
| | AFUB_099610 | A protein with a signal peptide and a Tricorn protease N-terminal domain (Superfamily SSF69304). |
| | AFUB_094790 | A disordered region and a region with basic and acidic residues are annotated in UniProt. |

[a]The description from FungiDB Release 68.

in teichoic acid biosynthesis. However, its involvement in the teichoic acid biosynthesis remains unclear. Using AlphaFold structure prediction in the AlphaFold Protein Structure Database of EMBL-EBI (23, 44), the structure of *A. fumigatus* FacC was clustered with the structure of bacterial glycosylases. FacC has a signal peptide in its N-terminal region that is cleaved between Ala17 and Lys18 (45), indicating that it is a secreted protein and plays a role in the extracellular space. A domain containing IRP048799 P68 RBP/TagC-like, a beta-propeller, and a PF21311 Phage 5-bladed beta-propeller receptor-binding platform domain was also found in this protein. This domain has been suggested to bind the N-acetylglucosamine moiety (46, 47). However, the relationship of this domain with azole resistance cannot be fully inferred from this information, so further studies using gene-deletion strains and other methods are necessary.

AihA (AFUB_046570) has been annotated as a *Porphyromonas*-type peptidyl arginine deiminase. Amino acid sequence comparisons using BLASTP (48) and a comparison of the structure predicted using AlphaFold and clustered using MMseq2 and FoldSeek (23, 44) showed that this protein is a putative member of the agmatine deiminase family. Moreover, guanidino-binding and catalytic residues (49) were found to be conserved in AihA (Fig. S5). Agmatine deiminases are broadly found in bacterial species, and more than 100 protein sequences have been identified and deposited in public databases. This enzyme catalyzes the deimination of agmatine, a decarboxylated product of arginine, into *N*-carbamoylputrescine and ammonia. Ammonium is known to act as a virulence factor among bacterial species, promoting the survival of pathogens in the host environment. Interestingly, γ-glutamyl transpeptidases also produce ammonia from the enzymatic reaction described above. Therefore, although ammonia might contribute to azole resistance, the underlying mechanisms remain unknown.

Information on AFUB_099610 and AFUB_094790 is limited because well-characterized domains were not detected in these proteins, and well-characterized proteins were not clustered with them. AFUB_099610 was annotated as a protein with a signal peptide and a tricorn protease N-terminal domain (superfamily SSF69304). The N-terminal region of AFUB_099610 contains a signal peptide that is cleaved between the residues Ala19 and Ala20. Tricorn protease is known as the core of a proteolytic system in *Thermoplasma acidophilum*, which degrades 7- to 9-residue-long peptides produced by the proteasome-mediated degradation of cellular proteins (50). This protease is a 121 kDa protein that forms a homohexamer (720 kDa). By contrast, the weight of AFUB_099610 is only 42 kDa. Because of the significant difference in protein size and the annotation of the N-terminal region as part of the superfamily, the actual function of AFUB_099610 remains unclear. The AFUB_094790 protein has two features: a disordered region and a region with basic and acidic residues. Considering the features of the disordered region, its predicted structure had a low confidence score (https://alphafold.ebi.ac.uk/entry/B0YD98). Although the C2H2 finger domain transcription factors CON7-like (InterPro IPR039327) and C2H2 FINGER DOMAIN TRANSCRIPTION FACTOR (EUROFUNG)-RELATED (PANTHER PTHR36167:SF3) were added to the protein as other annotations, this protein has no C2H2 domain.

This study had a few limitations. First, although the observed MIC shifts were relatively small (≤2 fold), such changes may still hold biological and clinical significance. As highlighted by Boyce (51), microevolutionary processes in pathogenic fungi can involve subtle, cumulative changes in drug resistance that ultimately contribute to the gradual emergence of antifungal resistance. These findings underscore the importance of not overlooking even minor MIC shifts, especially in the context of prolonged antifungal exposure. Another limitation is that such RttA mutations have not been identified in clinical isolates, and their effects have not been evaluated in any *in vivo* model, such as mouse models. Further investigations using infection models and expanded surveillance of clinical isolates are warranted to enhance the translational relevance of these findings. Several genes potentially regulated by RttA have been identified, but their functions remain largely unclear, as mentioned above. Moreover, some of these genes encode proteins with low homology, whose functions are

almost unknown. Therefore, further investigation is expected to elucidate the functional connection between RttA and azole antifungal resistance. In addition, sequences upstream of these genes, which were predicted to be RttA binding sites, need to be confirmed through experimental data such as ChIP-seq. Finally, the data were derived from the parental strain A1159 and its related derivatives. Therefore, further analyses using strains of different genetic backgrounds may help deepen our understanding. Clarifying these points will allow us to gain a more detailed understanding of the molecular function of RttA.

In summary, in this study, RttA was reannotated as a transcription factor containing a $Zn_2$-$Cys_6$ fungal-type DNA-binding domain. This protein contributes to azole resistance and is associated with expression changes in six genes that contain sequences similar to the UPC2-binding site in the promoter regions of these genes, although the precise functions of these genes on azole resistance remain unknown. These findings suggest that RttA is involved in regulating genes that contribute to azole resistance, highlighting its potential as a target for developing antifungal strategies against *A. fumigatus*.

## ACKNOWLEDGMENTS

This study was supported by AMED under grant numbers JP19fm0208024 and JP21jm0110015 and the Joint Usage/Research Program of the Medical Mycology Research Center, Chiba University (22-02 and 23-02). We also thank Kasumi Kodama for technical assistance. *A. fumigatus* A1159 and the pSK485 plasmid were obtained from the Fungal Genetics Stock Center (Manhattan, KS, USA).

## AUTHOR AFFILIATIONS

[1]Department of Pharmaceutical Sciences, School of Pharmacy at Narita, International University of Health and Welfare, Narita, Chiba Prefecture, Japan

[2]Department of Veterinary Medicine, Obihiro University of Agriculture and Veterinary Medicine, Obihiro, Hokkaido Prefecture, Japan

[3]Diagnostic Center for Animal Health and Food Safety, Obihiro University of Agriculture and Veterinary Medicine, Obihiro, Hokkaido Prefecture, Japan

[4]Medical Mycology Research Center, Chiba University, Chiba, Chiba Prefecture, Japan

[5]Department of Life and Environmental Sciences, University of Tsukuba, Tsukuba, Ibaraki Prefecture, Japan

[6]Microbiology Research Center for Sustainability (MiCS), University of Tsukuba, Tsukuba, Ibaraki Prefecture, Japan

## AUTHOR ORCIDs

Takahito Toyotome http://orcid.org/0000-0003-2822-2767
Hiroki Takahashi http://orcid.org/0000-0001-5627-1035
Akira Watanabe http://orcid.org/0000-0002-3057-2937
Daisuke Hagiwara http://orcid.org/0000-0003-1382-3914

## FUNDING

| Funder | Grant(s) | Author(s) |
| --- | --- | --- |
| Japan Agency for Medical Research and Development | JP19fm0208024 | Takahito Toyotome |
| | | Hiroki Takahashi |
| | | Akira Watanabe |
| | | Daisuke Hagiwara |
| Japan Agency for Medical Research and Development | JP21jm0110015 | Akira Watanabe |
| | | Daisuke Hagiwara |
| Joint Usage/Research Program of the Medical Mycology Research Center, Chiba University | 22-02 | Takahito Toyotome |

| Funder | Grant(s) | Author(s) |
|---|---|---|
| | | Akira Watanabe |
| Joint Usage/Research Program of the Medical Mycology Research Center, Chiba University | 23-02 | Takahito Toyotome |
| | | Akira Watanabe |

## AUTHOR CONTRIBUTIONS

Takahito Toyotome, Conceptualization, Data curation, Formal analysis, Funding acquisition, Investigation, Methodology, Project administration, Resources, Supervision, Validation, Visualization, Writing – original draft, Writing – review and editing | Hiroki Takahashi, Conceptualization, Funding acquisition, Investigation, Methodology, Supervision, Writing – original draft, Writing – review and editing, Project administration | Akira Watanabe, Conceptualization, Funding acquisition, Investigation, Methodology, Supervision, Writing – original draft, Writing – review and editing | Daisuke Hagiwara, Conceptualization, Funding acquisition, Investigation, Methodology, Supervision, Visualization, Writing – original draft, Writing – review and editing

## DATA AVAILABILITY

The sequence data obtained through next-generation sequencing were deposited in the rDNA Data Bank of Japan/ European Nucleotide Archive/Genetic Sequence Database (DDBJ/ENA/GenBank) under accession no. DRR640327 as the RNA-Seq data to confirm the coding region of *rttA* in the A1159 strain and accession no. DRR640324, DRR640325, and DRR640326 for the RNA-Seq data from A1159, *ΔrttA*, and *rttA*OE strains, respectively. For any additional data not presented in this study, please contact the authors directly.

## ADDITIONAL FILES

The following material is available online.

### Supplemental Material

**Figure S1 (Spectrum01810-25-s0001.TIF).** An overlooked exon in the rttA gene.
**Figure S2 (Spectrum01810-25-s0002.TIF).** The structure around rttA gene in the constructed ΔrttA and rttAOE strains.
**Figure S3 (Spectrum01810-25-s0003.TIF).** PCA plot based on RNA-Seq data from three Aspergillus fumigatus strains.
**Figure S4 (Spectrum01810-25-s0004.tiff).** Structure of the predicted *ggtA* gene.
**Figure S5 (Spectrum01810-25-s0005.TIF).** Sequence alignment of AihA from Aspergillus fumigatus (upper line), peptidylarginine deiminase from Porphyromonas gingivalis (middle line), and agmatine deiminase from Dyadobacter fermentans (lower line).
**Supplemental tables (Spectrum01810-25-s0006.xlsx).** Tables S1 to S11.

### Open Peer Review

**PEER REVIEW HISTORY (review-history.pdf).** An accounting of the reviewer comments and feedback.

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
