## [Reviewer comments · Microbiology Spectrum]

Microbiology Spectrum

RttA, a Zn₂-Cys₆ transcription factor in *Aspergillus fumigatus*, contributes to azole tolerance

Takahito Toyotome, Hiroki Takahashi, Akira Watanabe, and Daisuke Hagiwara

Corresponding Author(s): Takahito Toyotome, Kokusai Iryo Fukushi Daigaku - Narita Campus

Review Timeline:

Submission Date:	July 10, 2025
Editorial Decision:	July 14, 2025
Revision Received:	July 16, 2025
Accepted:	July 17, 2025

Editor: Gustavo Goldman

Reviewer(s): The reviewers have opted to remain anonymous.

Transaction Report:

DOI: <https://doi.org/10.1128/spectrum.01810-25>

Re: Spectrum01810-25 (RttA, a Zn₂-Cys₆ transcription factor in *Aspergillus fumigatus*, contributes to azole tolerance)

Dear Dr. Takahito Toyotome:

Thank you for the privilege of reviewing your work. Below you will find my comments, instructions from the Spectrum editorial office, and the reviewer comments.

The authors have addressed all the comments and suggestions of the reviewers, and the manuscript is ready for acceptance. I only suggest, when it is possible, that the word "tolerance" be replaced throughout the manuscript by "resistance" to avoid possible misinterpretations about resistance versus tolerance/persistence.

Revision Guidelines

Sincerely,
Gustavo Goldman
Editor
Microbiology Spectrum

Dear Editor,

We sincerely thank you for the constructive suggestion regarding our manuscript.

Below, we provide our response to the comment:

Comment:

The authors have addressed all the comments and suggestions of the reviewers, and the manuscript is ready for acceptance. I only suggest, when it is possible, that the word "tolerance" be replaced throughout the manuscript by "resistance" to avoid possible misinterpretations about resistance versus tolerance/persistence.

Response:

We appreciate this valuable suggestion. Accordingly, we have carefully revised the manuscript and replaced the term "tolerance" with "resistance" throughout the text to avoid potential misinterpretation. We believe this change improves the clarity and accuracy of the manuscript.

We hope the revised version meets the requirements for final acceptance.

Re: Spectrum01810-25R1 (RttA, a Zn₂-Cys₆ transcription factor in *Aspergillus fumigatus*, contributes to azole tolerance)

Dear Dr. Takahito Toyotome:

Manuscript ready for acceptabce

Your manuscript has been accepted, and I am forwarding it to the ASM production staff for publication. Your paper will first be checked to make sure all elements meet the technical requirements. ASM staff will contact you if anything needs to be revised before copyediting and production can begin. Otherwise, you will be notified when your proofs are ready to be viewed.

Sincerely,
Gustavo Goldman
Editor
Microbiology Spectrum